# Prevalence of Electrocardiographic Abnormalities in Patients with Acute Pulmonary Embolism: A Systematic Review and Meta-Analysis

**DOI:** 10.3390/jcm14134750

**Published:** 2025-07-04

**Authors:** Sarunsorn Krintratun, Wuttipong Srichuachom, Wachira Wongtanasarasin

**Affiliations:** 1Department of Emergency Medicine, Faculty of Medicine, Chiang Mai University, Chiang Mai 50200, Thailand; sarunsorn.k@cmu.ac.th (S.K.); wuttipong.sr@cmu.ac.th (W.S.); 2Acute Care and Emergency Medicine (ACE) Research Cluster, Faculty of Medicine, Chiang Mai University, Chiang Mai 50200, Thailand

**Keywords:** pulmonary embolism, electrocardiogram, S1Q3T3, prevalence, sinus tachycardia, systematic review

## Abstract

**Background/Objectives**: Acute pulmonary embolism (PE) remains a leading cause of cardiovascular morbidity and mortality. Although computed tomography pulmonary angiography (CTPA) is the gold standard for diagnosis, electrocardiography (ECG) is a widely available, non-invasive tool that may provide diagnostic clues. This study aims to estimate the pooled prevalence of specific ECG abnormalities in patients with confirmed acute PE. **Methods**: We conducted a systematic review and meta-analysis in accordance with the PRISMA guidelines. We searched PubMed, Embase, Web of Science, Scopus, and the Cochrane Central Register of Controlled Trials until April 2024 for studies reporting prevalence data on ECG abnormalities in confirmed acute PE cases. Pooled prevalence estimates were calculated using a random-effects model, and heterogeneity was assessed using the *I*^2^ statistic. Publication bias was evaluated through funnel plots and Egger’s test. **Results**: Twenty-four studies with 7467 patients were included. The most common ECG abnormalities were sinus tachycardia (31%, 95% CI 22–40%), clockwise rotation (28%, 95% CI 12–45%), T-wave inversion in leads V1–V3 (18%, 95% CI 13–23%), S1Q3T3 pattern (15%, 95% CI 11–19%), and right bundle branch block (14%, 95% CI 10–17%). High heterogeneity was observed across studies, with an *I*^2^ value exceeding 95%. Publication bias was detected for both S1Q3T3 and right bundle branch block. **Conclusions**: Sinus tachycardia and the S1Q3T3 pattern are frequently observed in acute PE, supporting their potential use in clinical recognition. However, significant heterogeneity and publication bias highlight the need for larger, higher-quality studies with standardized ECG protocols to understand ECG’s diagnostic and prognostic role in PE.

## 1. Introduction

Acute pulmonary embolism (PE) is a significant cause of cardiovascular morbidity and mortality, with an estimated incidence of 60 to 124 cases per 100,000 persons [1,2]. Despite advances in diagnostic imaging, mortality remains high, ranging from 8% to 30%, particularly in cases complicated by diagnostic delay, hemodynamic instability, or underlying comorbid conditions [1].

The clinical presentation of acute PE is notoriously variable, often mimicking other cardiopulmonary disorders [3]. Symptoms may include dyspnea, pleuritic chest pain, syncope, hemoptysis, or may be absent [3,4]. This diagnostic ambiguity contributes to the condition’s under-recognition and underscores the need for adjunctive tools to guide early clinical suspicion. While computed tomography pulmonary angiography (CTPA) is the diagnostic gold standard, its use is limited by availability, radiation exposure, and contrast-related risks [5]. Electrocardiography (ECG), on the other hand, is a universally accessible, rapid, and non-invasive diagnostic modality. Although it lacks the sensitivity and specificity to serve as a definitive diagnostic test, certain ECG patterns—such as sinus tachycardia, the S1Q3T3 pattern, right bundle branch block (RBBB), and T-wave inversions in precordial leads—have been associated with right ventricular strain in the context of PE [6,7]. These findings may offer valuable clinical clues, particularly when advanced imaging is delayed or unavailable [6].

Several recent studies have reported additional ECG abnormalities in acute PE, including right-axis deviation, low voltage, ST segment changes, and atrial arrhythmias [8,9]. While individual studies suggest the potential diagnostic and prognostic utility of these patterns, prevalence estimates vary widely, and their role in risk stratification remains uncertain.

To date, no comprehensive synthesis of the prevalence of ECG findings in confirmed cases of acute PE has been undertaken with sufficient statistical rigor. We therefore conducted a systematic review and meta-analysis to determine the pooled prevalence of specific ECG abnormalities in patients with imaging-confirmed acute PE. The findings aim to inform clinicians on the diagnostic utility of ECG patterns and support the development of improved early recognition strategies.

## 2. Materials and Methods

### 2.1. Study Design and Registration

We conducted a systematic review and meta-analysis to estimate the prevalence of ECG findings in patients with confirmed acute PE. The study adhered to the Preferred Reporting Items for Systematic Reviews and Meta-Analyses guidelines [10]. The protocol was prospectively registered with the International Prospective Register of Systematic Reviews (PROSPERO; registration ID: CRD42024535571). We used ChatGPT 4.0 and Grammarly to check and correct grammatical errors during the writing of this document. After using these tools, we reviewed, checked, and edited the content as needed, taking full responsibility for its accuracy.

### 2.2. Search Strategy and Study Selection

A comprehensive literature search was conducted across five databases—PubMed, Embase, Web of Science, Scopus, and the Cochrane Central Register of Controlled Trials—from inception to 8 April 2024. We used a combination of the Medical Subject Heading (MeSH) terms with various spellings and endings: “prevalence,” “epidemiology,” “incidence,” “finding,” “epidemiology,” “pulmonary embolism,” “pulmonary thromboembolic disease,” “electrocardiography,” “electrocardiogram,” “ECG,” and “EKG.” The complete search strategy is provided in Appendix A. No language restrictions were applied. Additional studies were identified through reference screening and citation tracking of eligible articles and relevant reviews. We extracted the search results from these databases and deleted the duplicates. The remaining articles were added to the Rayyan website for screening.

### 2.3. Inclusion Criteria and Outcome of Interest

Studies were included if they met the following criteria: (1) patients had confirmed acute PE diagnosed by CTPA, ventilation-perfusion (V/Q) scanning, or pulmonary angiography; (2) ECG findings were systematically reported using 12-lead electrocardiography; and (3) the study provided prevalence data on specific ECG abnormalities. We excluded preclinical studies, case reports, case series, narrative reviews, editorials, and studies without sufficient ECG detail. Only studies published in English were included in the final analysis.

The primary endpoint is the prevalence of ECG abnormalities in confirmed acute PE cases. ECG patterns included the S1Q3T3 pattern, the S1S2S3 pattern, RBBB, right-axis deviation, clockwise rotation, sinus tachycardia, atrial arrhythmia, atrial fibrillation and atrial flutter, low voltage, P pulmonale, and T-wave inversion in precordial leads (leads V1–V3). These abnormalities were selected as mentioned in the standard international guidelines (i.e., European Society of Cardiology 2019) [5].

### 2.4. Data Extraction and Study Risk of Bias Assessment

Titles and abstracts were independently screened by two reviewers (W.W. and S.K.) using Rayyan (Qatar Computing Research Institute), followed by full-text review of eligible articles. Disagreements were resolved through discussion or by a third person (W.S.).

A standardized data extraction form was used to collect the following variables: first author, publication year, study design, country, enrollment period, sample size, patient demographics, PE confirmation method, and reported ECG findings. All extracted data were entered into Microsoft Excel. Two reviewers (S.K. and W.S.) independently assessed the methodological quality of the included studies using the Joanna Briggs Institute (JBI) critical appraisal checklist for prevalence studies [11]. Discrepancies were resolved through consensus, with the assistance of a third person (W.W.). Studies were categorized as having low risk of bias (score 8–9), some concerns (score 5–7), or high risk of bias (score < 5).

### 2.5. Statistical Analysis

We calculated pooled prevalence estimates and 95% confidence intervals (CIs) for each ECG finding using a random-effects model (DerSimonian and Laird method) to account for between-study heterogeneity. Heterogeneity was assessed using the *I*^2^ statistic and Cochran’s Q test, with *I*^2^ values of <25%, 25–50%, and >50% interpreted as low, moderate, and high heterogeneity, respectively [12]. Publication bias was assessed visually using funnel plots and statistically using Begg’s and Egger’s tests. All tests were two-sided, with a *p*-value of less than 0.05, which was considered statistically significant. All analyses were conducted using STATA MP version 16.0 (StataCorp LLC, College Station, TX, USA).

## 3. Results

### 3.1. Study Selection and Characteristics of Included Studies

The literature search yielded 1715 records. After removing duplicates, 1363 studies underwent title and abstract screening, of which 61 full-text articles were assessed for eligibility. A total of 24 studies, comprising 7467 patients with confirmed acute PE, met the inclusion criteria and were included in the final analysis [7,9,13,14,15,16,17,18,19,20,21,22,23,24,25,26,27,28,29,30,31,32,33,34] (Figure 1, PRISMA flow diagram).

The included studies were published between 2000 and 2024 and conducted across 13 countries. Sample sizes ranged from 20 to 1676 patients. All studies confirmed the diagnosis of acute PE via CTPA or ventilation-perfusion scan. Study characteristics and patient demographics are described in Table 1. Fourteen studies (58.3%) were classified as low risk (JBI scores 8–9), and ten studies (41.7%) were considered to have some concerns (JBI scores 5–7). Most concerns arose from unclear sampling methods, inadequate sample size, the absence of consecutive patient inclusion, and a lack of standardized ECG interpretation. Details of the risk of bias for each included study are listed in Appendix A.

### 3.2. Prevalence of Electrocardiographic Abnormalities

Eleven distinct ECG abnormalities were identified, and their pooled prevalence rates were estimated. Figure 2 summarizes each abnormality’s prevalence and 95% CIs. The prevalence of the S1Q3T3 pattern was reported in all included studies [7,9,13,14,15,16,17,18,19,20,21,22,23,24,25,26,27,28,29,30,31,32,33,34], followed by RBBB (21 studies) [7,9,13,14,16,17,18,20,21,22,23,24,25,26,28,29,30,31,32,33,34], right axis deviation (13 studies) [9,13,14,17,20,21,24,25,26,27,30,31,33], and sinus tachycardia (13 studies) [7,13,14,16,17,18,19,21,27,29,31,33,34].

The pooled prevalence of the S1Q3T3 pattern was 15% (95% CI, 11–19%; *I*^2^ = 97%, *p* < 0.001) across 24 studies (Appendix A). Among low-risk studies, the prevalence was 15% (95% CI, 9–20%; *I*^2^ = 98%), while studies with some concerns showed a slightly higher pooled prevalence of 16% (95% CI, 9–23%; *I*^2^ = 95%). No significant difference in prevalence was observed between risk groups (*p* = 0.748).

Sinus tachycardia was found to be the most prevalent finding in 31% of patients (95% CI, 22–40%; *I*^2^ = 97%, *p* < 0.001) based on data from 13 studies (Appendix A). The pooled prevalence from low-risk studies was 34% (95% CI, 21–48%), compared with 27% (95% CI, 23–30%) in studies with some concerns. The difference in pooled estimates between risk groups was statistically significant (*p* = 0.286).

The details regarding the prevalence and corresponding 95% CIs of other ECG abnormalities are summarized and reported in the Appendix A.

Visual inspection of the funnel plots for the two most reported ECG findings (S1Q3T3 and RBBB) revealed mild asymmetry (Appendix A), particularly among smaller studies. Egger’s test indicated publication bias for both findings (*p* = 0.0238 and *p* = 0.0239, respectively).

## 4. Discussion

The most frequently observed ECG abnormalities in acute PE include sinus tachycardia, clockwise rotation, T-wave inversions in precordial leads, S1Q3T3 pattern, and RBBB. These findings are consistent with prior literature and support the clinical relevance of certain ECG features as markers of right ventricular strain in the setting of acute PE [35].

Among these, sinus tachycardia demonstrated the highest pooled prevalence (31%), consistent with its frequent inclusion in clinical prediction rules such as the Wells and revised Geneva scores [36,37]. While non-specific, sinus tachycardia may reflect a compensatory response to impaired hemodynamics, possibly mediated by the release of catecholamine and inflammatory mediators [8,34]. Conversely, the S1Q3T3 pattern, also known as the McGinn–White sign, traditionally regarded as a hallmark of acute PE, was found in 15% of cases, aligning with its high specificity but low sensitivity [33]. This pattern likely reflects acute right ventricular overload and a shift in the cardiac electrical axis due to pulmonary arterial obstruction.

RBBB and right-axis deviation, each associated with elevated right ventricular pressures and dilation, were also commonly observed [31,32,38]. Transient RBBB, in particular, has been reported as reversible with anticoagulation therapy, suggesting a direct hemodynamic effect [32]. Clockwise rotation and delayed R/S transition (V5 or beyond) similarly reflect right ventricular dilatation and conduction delay [8,34]. Notably, these patterns have demonstrated high specificity, and some studies suggest associations with adverse outcomes, including short-term mortality [32].

T-wave inversions in leads V1–V3, found in a significant proportion of patients, are thought to result from subendocardial ischemia of the right ventricle, often secondary to increased wall tension and impaired perfusion [39]. Compared with S1Q3T3 or RBBB, T-wave inversion may offer higher sensitivity and moderate diagnostic accuracy, underscoring its potential diagnostic value in combination with other findings [33,40].

Less prevalent but noteworthy findings included atrial arrhythmias (such as atrial fibrillation and flutter), low-voltage QRS complexes, and the S1S2S3 pattern. Atrial arrhythmias may result from atrial stretch and hypoxia-induced ectopic activity, and their presence may indicate more advanced cardiopulmonary compromise [41]. Although the S1S2S3 pattern is historically linked to right ventricular hypertrophy and chronic pulmonary disease [42], its occurrence in acute PE remains controversial, with some studies suggesting its association with right heart strain and others finding no significant correlation [30].

The pathophysiological basis of these ECG changes is largely attributed to acute pressure overload and right ventricular dilatation, leading to altered depolarization vectors and conduction abnormalities [26], as summarized in Table 2. Such changes may occur transiently and are often reversible following appropriate treatment. Moreover, accumulating evidence suggests that certain ECG features—particularly RBBB and T-wave inversions—may also carry prognostic significance in risk stratification tools, potentially informing decisions regarding thrombolysis or intensive monitoring [43,44,45].

### 4.1. Strengths and Limitations

This meta-analysis incorporates data from a large international cohort and applies rigorous methodological standards, including pre-registration and risk-of-bias assessment using the JBI checklist [11]. The study also includes a broad spectrum of ECG abnormalities, reflecting the heterogeneous manifestations of PE on ECG. Nevertheless, several limitations merit consideration. First, although we employed rigorous methods to identify and include relevant studies, the overall quality and design of the included studies were variable. Some studies had small sample sizes, lacked uniform diagnostic criteria, or used inconsistent ECG interpretation protocols, which may have introduced bias and affected the reliability of pooled estimates. Future prospective studies with standardized ECG acquisition and interpretation protocols are needed to explore the co-occurrence and temporal dynamics of these findings. Second, heterogeneity across studies was substantial, likely attributable to differences in study design, populations, and timing of ECG acquisition relative to PE diagnosis. Including data from a wide range of geographical regions, age groups, and sexes may have further contributed to this heterogeneity, reflecting variations in healthcare systems, baseline characteristics, and diagnostic practices. Third, incomplete reporting in the primary studies precluded subgroup analyses by sex, age, PE severity, or comorbidities. Additionally, the lack of standardized ECG acquisition protocols and variability in outcome definitions may have affected study comparability. Finally, evidence of publication bias was identified for some findings, particularly among smaller studies, which may influence the robustness of the pooled estimates.

### 4.2. Clinical Implications and Future Directions

ECG remains a widely accessible and cost-effective tool, and although it is insufficiently sensitive to rule out PE, it may raise clinical suspicion in appropriate contexts. However, International guidelines such as those of the European Society of Cardiology (ESC, 2019) [5] reinforce that the ECG should not replace imaging tests, such as CTPA, which is recommended as the gold standard. Similarly, several reports published earlier mentioned the risk of delays in diagnosis if there is excessive reliance on the ECG [46,47,48]. Furthermore, the initial diagnosis of PE should begin with a structured clinical evaluation, using pre-test probability scores, such as the Wells score or the Geneva score, followed by D-dimer measurement in cases of low or moderate probability. Patients with signs of hemodynamic instability should undergo CTPA immediately, the gold standard examination for diagnostic confirmation. In settings with limited resources or contraindications to CTPA, V/Q scintigraphy and bedside echocardiography can be used. Given the observed prevalence of right heart strain patterns, future studies should explore the additive value of ECG findings in diagnostic algorithms and risk stratification scores. Furthermore, given the inherent overlap of ECG findings, combining a group of findings (i.e., sinus tachycardia and right-axis deviation) would be beneficial to address a characteristic group of ECG abnormalities in patients with acute PE. Further prospective research is also needed to evaluate the temporal evolution of ECG changes in PE and their prognostic significance across varying clinical severities.

## 5. Conclusions

In conclusion, this systematic review and meta-analysis highlight the high prevalence of several key electrocardiographic abnormalities associated with acute PE, with sinus tachycardia and clockwise rotation being the most frequently observed. These findings underscore the potential diagnostic value of ECG as an accessible and non-invasive tool for clinicians, particularly in settings where advanced imaging may be unavailable or delayed. Identifying right heart strain patterns, such as RBBB and T-wave inversions in precordial leads, further supports the ECG’s role in raising clinical suspicion and guiding early recognition of PE. However, given the variability in prevalence estimates across studies, the need for standardized ECG interpretation and larger, high-quality studies remains critical. Future research should focus on the prognostic significance of these ECG findings and their integration into risk stratification models to enhance patient management. Ultimately, this study provides a foundation for further exploration of ECG’s utility in the early diagnosis and management of acute PE.

## Figures and Tables

**Figure 1 jcm-14-04750-f001:**
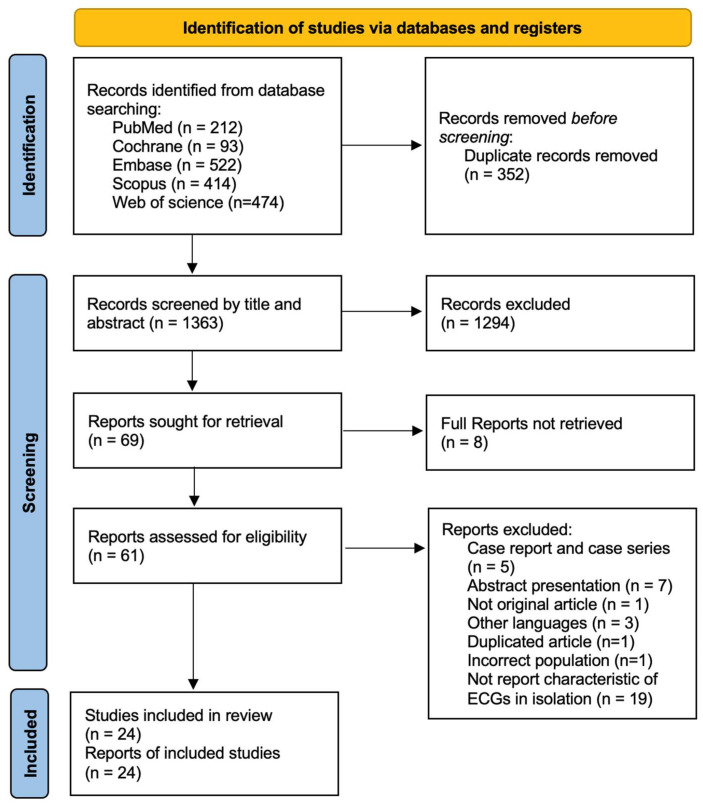
PRISMA flow diagram.

**Figure 2 jcm-14-04750-f002:**
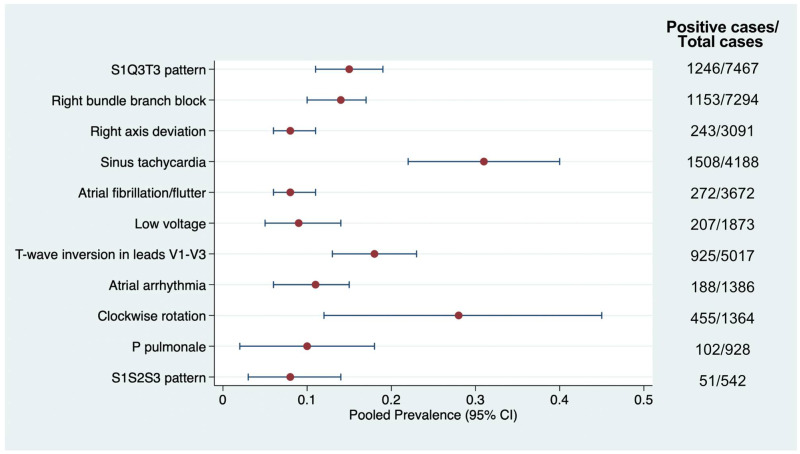
Pooled prevalence of ECG abnormalities in confirmed acute PE cases.

**Figure 3 jcm-14-04750-f003:**
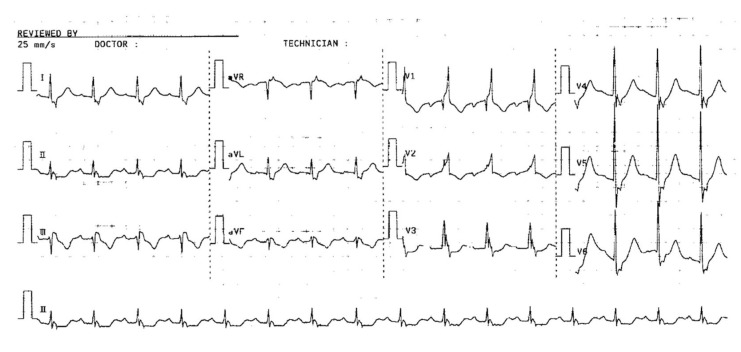
12-lead electrocardiogram shows right bundle branch block, T wave inversion in leads V1–V2, and S1Q3T3 pattern (Case courtesy: Dr. Wachira Wongtanasarasin).

**Table 1 jcm-14-04750-t001:** Characteristics of included studies.

First Author, Year	Country	*n*	Age (Year), mean ± SD	%Male	Design	Inclusion Criteria	ECG Findings of Interest
Wang, 2023 [14]	China	341	69 ± 14	50.4	Multicenter retrospective cohort study	- >18 years - Confirmed acute PE by multidetector CTPA	- S1Q3T3 pattern - Arrhythmias and bundle branch block - Axis and chamber enlargement - T wave inversion in V1–V3
Kusayama, 2019 [27]	Japan	65	67 ± 13	32.1	Retrospective chart review study	- Confirmed acute PE by contrast-enhanced CT and/or pulmonary angiography	- Arrhythmias and bundle branch block - Axis and chamber enlargement - S1S2S3 pattern and S1Q3T3 pattern - ST segment abnormalities
Pourafkari, 2017 [25]	United States	200	61 ± 17	52.5	Retrospective observational cohort	- Confirmed acute PE by CTPA - Received 12-lead ECG	- Arrhythmias and bundle branch block - S1Q3T3 pattern - Axis and chamber enlargement - QRS wave in V1 - ST elevation in lead aVR
Çagdas, 2018 [30]	Turkey	106	60 ± 18	50.0	Retrospective observational cohort	- Confirmed acute PE by CTPA	- Arrhythmias and bundle branch block - Axis and chamber enlargement - T wave inversion in precordial lead - ST segment abnormalities - S1Q3T3 pattern and S1S2S3 pattern
Park, 2017 [18]	South Korea	89	68 ± 15	59.5%	Retrospective study	- Confirmed acute PE by CTPA	- Arrhythmias and bundle branch block - S1Q3T3 pattern - Low voltage - T wave inversion in lead V1–V3
Rodger, 2000 [13]	Canada	49	N/A	N/A	Retrospective observational case-control	- Acute PE confirmed by ventilation perfusion scan or pulmonary angiogram	- Arrhythmias and bundle branch block - Axis and chamber enlargement - S1Q3T3 pattern - S slurred in V1 and V2 - S1S2S3 pattern - ST segment abnormalities - T wave inversion in lead III and aVF - T wave inversion in lead V1 or V2 - T flat or inverted in any lead - T flat or inverted in any lead but aVR
Bahreini, 2024 [9]	Iran	250	54 ± 19	60.0	Prospective cross-sectional study	- Diagnosed acute PE by using specific ICD-10 codes for acute PE	- Axis and chamber enlargement - Arrhythmias and bundle branch block - ST segment abnormalities - S1Q3T3 pattern - T wave inversion in lead V1-V6 - QTc prolongation
Thomson, 2019 [33]	Scotland	189	66 (range 20–93)	43.4	Retrospective case-control study	- Presented D-dimer and CTPA in confirmed acute PE	- Arrhythmias and bundle branch block - T wave inversion in lead II, III, aVF, V1-V4 - Axis and chamber enlargement - S1Q3T3 pattern
Zhan, 2014 [15]	China	20	58 ± 10	40	Retrospective study	- Signs and symptoms suggesting acute PE - Hemodynamic stability - Acute PE confirmed by CTPA - Available ECG of good technical quality - No obvious history of cardiopulmonary disease	- S1Q3 pattern and S1Q3T3 pattern - T wave inversion in lead V2-V4 - T wave inversion in III and aVF - QRS wave in V1 - ST segment abnormalities
Weekes, 2022 [16]	United States	1676	60 ± 17	51.8	Prospective, observational, multicenter cohort study	- Confirmed acute PE by contrast-enhanced chest CT or detection of high probability on nuclear ventilation perfusion scan within 12 h of ED administration	- Arrhythmias and bundle branch block - S1Q3T3 pattern - ST elevation in lead V1 - T wave inversion in lead V2-V4 - T wave inversion in lead II, III, and aVF - ST segment abnormalities - Axis and chamber enlargement
Richman, 2004 [29]	United States	49	69 (no SD reported)	NI	Retrospective case-control study	- Confirmed acute PE by CT protocol to rule out thromboembolic disease	- Arrhythmias and bundle branch block - Axis and chamber enlargement - ST segment abnormalities - Peaked, biphasic, and inverted T wave - S1Q3T3 pattern
Kukla, 2011 [26]	Poland	292	65 ± 15	37.3	Retrospective study	- Confirmed acute PE by CT, ECG, Doppler ultrasound of the proximal deep veins of the lower extremity, and scintigraphy	- Arrhythmias and bundle branch block - Axis and chamber enlargement - S1Q3T3 pattern - T wave inversion in lead V2-V4 - ST segment abnormalities - QR sign in lead V1
Novicic, 2020 [32]	Serbia	110	65 ± 13	45.5	Retrospective study	- Confirmed acute PE by CTPA with intermediate-high and high-risk PE - Good quality ECG record	- S1Q3T3 pattern - S1 deep in lead I - S wave in aVL - T wave inversion in lead V1-V6 - Bundle branch block
Cetin, 2016 [24]	Turkey	249	66 ± 16	33.8	Prospective study	- Confirmed acute PE by CTPA	- Arrhythmias and bundle branch block - S1Q3T3 pattern - Axis and chamber enlargement
Obradovic, 2016 [28]	Serbia	144	60 ± 16	50	Ambispective study	- Diagnosed acute PE by MDCT-PA	- S1Q3T3 pattern - Arrhythmias and bundle branch block - T wave inversion in lead V1-V6 - S wave in aVL
Ivan, 2017 [8]	United States	352	68 (range 23-96)	37.5	Retrospective study	- ED diagnosed acute PE by CTA and VQ scan	- Arrhythmias and bundle branch block - Axis and chamber enlargement - S1Q3T3 pattern - T wave inversion and T wave flattening - ST depression
Casazza, 2018 [23]	Italy	1194	70 ± 16	42.8	Prospective multicenter study	- ≥18 years of age - Confirmed PE by CTPA, positive perfusion lung scan, pulmonary angiogram, combination of deep venous thrombosis at ultrasound of the lower or upper extremities in association with right ventricular dysfunction on an echocardiogram, detection of free-floating thrombi in the right atrium at echocardiography, or autopsy	- Arrhythmias and bundle branch block - S1Q3 pattern - T wave inversion in lead V1–V3 - ST elevation in inferior lead - QR sign in lead V1
Yan, 2024 [20]	China	383	67 ± 13	43	Prospective observational cohort study	- Adult patients with confirmed acute PE by CTPA - Non-high-risk classification - Admitted to the general wards - A standard 12-lead ECG at admission	- Arrhythmias and bundle branch block - S1Q3 pattern and S1Q3T3 pattern - T wave inversion in lead V1–V3 - ST elevation in inferior leads - QR sign in lead V1 - Axis and chamber enlargement
Vanni, 2009 [22]	Italy	386	67 ± 16	40	Prospective study	- Diagnosed acute PE by perfusion lung scan and spiral CT	- Bundle branch block - S1Q3T3 pattern - T wave inversion in lead V1–V3
Bolt, 2019 [21]	Switzerland	390	74 (69–81) ^1^	54	Prospective multicenter cohort study	- Consecutive patients aged ≥65 years - Objectively diagnosed acute PE - ECG performed within 24 h before or after the diagnosis of acute PE	- Arrhythmias and bundle branch block - Axis and chamber enlargement - S1Q3T3 pattern - QR sign in lead V1 - T wave inversion in lead V1-V4 - ST elevation in lead V1, aVR, or III - T wave inversion ≥ 3 leads
Geibel, 2005 [7]	Germany	508	63 ± 15	42	Prospective study	- Clinical suspicion of acute PE - Confirmation of acute PE by imaging (pulmonary angiography, ventilation perfusion lung scan, and leg ultrasound or phlebography)	- Arrhythmias and bundle branch block - S1Q3T3 pattern - Shift in transition zone to V5 - Q in lead III, aVF - ST segment abnormalities - T wave inversion in lead V2-V6 - Peripheral low voltage
Stein, 2013 [31]	United States	289	61 ± 18	44	Review medical records of hospitalized patients	- Aged ≥18 years - Diagnosed acute PE by CTPA - ECG obtained on the same day as diagnosed	- Arrhythmias and bundle branch block - Axis and chamber enlargement - S1S2S3 pattern and S1Q3T3 pattern
Witting, 2012 [19]	United States	97	50 ± 15	NI	Retrospective case-control study	- Diagnosed acute PE by CTPA	- T wave inversion in lead III, aVF, V1-2 - S1Q3T3 pattern and S1S2S3 pattern - Arrhythmias and bundle branch block - RV strain - Axis and chamber enlargement
Zhang, 2016 [34]	China	147	56 ± 15	41.5	Retrospective observational cohort	- Diagnosed acute PE by CTPA - ECG and CTPA were performed within 24 h following the onset of PE	- Arrhythmias and bundle branch block - Right ventricular hypertrophy - S1Q3T3 pattern - All RBBB - T wave inversion in lead V1–V3/V6, III, aVF - ST segment abnormalities - Peripheral low voltage - Prolonged QTc interval - PR segment depression in lead V1

^1^ Data was reported as median (interquartile range). Abbreviations: CT: computed tomography; CTPA: computed tomography pulmonary angiography; ECG: electrocardiogram; ED: emergency department; PE: pulmonary embolism; SD: standard deviation; NI: no information; RBBB: right bundle branch block.

**Table 2 jcm-14-04750-t002:** Pathophysiology and electrocardiogram findings in acute pulmonary embolism.

ECG Finding	Pathophysiology	ECG Characteristics
S1Q3T3 pattern (Figure 3)	Acute RV strain → altered conduction and repolarization	S wave in lead I, Q wave, and inverted T wave in lead III
Right bundle branch block (Figure 3)	RV dilation → delayed right ventricular conduction	RSR’ in V1, wide QRS > 120 ms, terminal S in I and V6
Right axis deviation	RV pressure overload or hypertrophy	QRS axis > +90°, dominant S in I and R in III
Sinus tachycardia	Sympathetic activation due to hypoxia, pain, or RV dysfunction	Regular rhythm, HR >100 bpm, normal P waves
Atrial fibrillation/Atrial flutter	Atrial strain or ischemia, particularly of the right atrium	Irregularly irregular rhythm (AF); flutter waves with sawtooth pattern (AFL)
Low voltage	Pericardial effusion, obesity, or shock state causing diminished signal	QRS < 5 mm in limb leads or <10 mm in precordial leads
T-wave inversion in leads V1–V3 (Figure 3)	Subendocardial ischemia of the RV due to pressure overload	Symmetrical T-wave inversion in V1–V3, sometimes extending to V4–V6
Atrial arrhythmia (other)	Right atrial enlargement or hypoxia-related automaticity	Ectopic P waves, PACs, atrial tachycardia
Clockwise rotation	RV enlargement causing altered depolarization vectors	R/S transition delayed beyond V4–V5, persistent S waves in V1–V4

Abbreviations: AF: Atrial fibrillation; AFL: Atrial flutter; ECG: electrocardiogram; HR: heart rate; PAC: premature atrial contraction; RV: right ventricle.

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
