# Peer review of "Prevalence of Electrocardiographic Abnormalities in Patients with Acute Pulmonary Embolism: A Systematic Review and Meta-Analysis"

_jcm, 2025, doi:10.3390/jcm14134750_

Round 1

Reviewer 1 Report

Comments and Suggestions for Authors

GENERAL COMMENTS: This is a systematic review and meta-analysis of ECG changes in patients with acute pulmonary embolism.  It represents a comprehensive search of existing literature, and results have wide appeal to all providers.  Some additional information would further enhance their manuscript.

SPECIFIC COMMENTS:

Figure 2:  Please add the number of patients represented by the number of studies in this pooled prevalence chart.

ECG patterns: The ECG changes described are dynamic and may be transient in appearance.  Several of rhythms listed may occur in the same patient at the same time (such as sinus tachycardia, right axis deviation and right bundle branch block) and thus the rhythms are not mutually exclusive.  It is not clear if information on the simultaneous appearance of multiple abnormalities can be ascertained from their review, but it deserves comment.

Constellation of ECG patterns:  Given the inherent overlap of ECG findings, was it possible to group ECG patterns to create a pooled prevalence of rhythms?  As noted above, sinus tachycardia and right axis deviation may represents a characteristic group of ECG abnormalities in acute pulmonary embolism.

ECG tracings: in a review of ECG abnormalities, actual tracings would enhance the manuscript. Specifically the tracings would be of more unusual or unique findings in acute pulmonary embolism such as S1Q3T3, clockwise rotation or low voltage findings. One tracing may be in the manuscript with the others in the supplement.

Author Response

Response to Reviewers

Date: 17 June 2025

Title: Prevalence of Electrocardiographic Abnormalities in Patients with Acute Pulmonary Embolism: A Systematic Review and Meta-analysis

Reviewer 1:

Comments

Responses

Correction on/
Mentioned on

GENERAL COMMENTS: This is a systematic review and meta-analysis of ECG changes in patients with acute pulmonary embolism.  It represents a comprehensive search of existing literature, and results have wide appeal to all providers.  Some additional information would further enhance their manuscript.

We appreciate the reviewer's positive feedback regarding the comprehensiveness of our literature search and the broad appeal of our findings.

SPECIFIC COMMENTS:

Figure 2:  Please add the number of patients represented by the number of studies in this pooled prevalence chart.

We agree with this valuable suggestion. Figure 2 displays the pooled prevalence estimates derived from our meta-analysis, which included 24 studies comprising a total of 7,467 patients. We revised the legend of Figure 2 to clearly state the number of positive and total cases for each ECG finding, providing a more complete context for the pooled prevalence values.

Figure 2

ECG patterns: The ECG changes described are dynamic and may be transient in appearance.  Several of rhythms listed may occur in the same patient at the same time (such as sinus tachycardia, right axis deviation and right bundle branch block) and thus the rhythms are not mutually exclusive.  It is not clear if information on the simultaneous appearance of multiple abnormalities can be ascertained from their review, but it deserves comment.

We concur with the reviewer's observation regarding the dynamic and often transient nature of ECG changes in acute PE, as well as the potential for multiple abnormalities to coexist simultaneously. Our manuscript acknowledges that ECG findings are attributed to acute pressure overload and right ventricular dilation and can be transient. Our meta-analysis focused on estimating the pooled prevalence of individual ECG abnormalities as reported in the included studies. Due to the retrospective nature and variability in reporting across the primary studies, information on the simultaneous appearance or evolution of multiple ECG patterns within the same patient could not be reliably ascertained or analyzed in our review. We have highlighted the "lack of standardized ECG interpretation criteria" as a limitation in our discussion. We’ve also added a comment in the Discussion section to further address this point, emphasizing that future prospective studies with standardized ECG acquisition and interpretation protocols are needed to explore the co-occurrence and temporal dynamics of these findings.

Page 11, Lines 215-218 and 231-238

Constellation of ECG patterns:  Given the inherent overlap of ECG findings, was it possible to group ECG patterns to create a pooled prevalence of rhythms?  As noted above, sinus tachycardia and right axis deviation may represents a characteristic group of ECG abnormalities in acute pulmonary embolism.

We acknowledge the valuable suggestion to explore the pooled prevalence of "constellations" or characteristic groups of ECG abnormalities. While this approach could provide deeper insights into the complex ECG manifestations of acute PE, our systematic review and meta-analysis primarily focused on the pooled prevalence of individual specific ECG abnormalities, as these were consistently reported across the included studies. The inherent variability in how ECG patterns were defined and reported in the heterogeneous primary studies, combined with the absence of standardized interpretation criteria, limited our ability to perform robust analyses of combined ECG patterns or specific constellations. We agree that identifying such characteristic groupings would be highly beneficial for clinical practice and will emphasize this as an important direction for future research in the "Clinical Implications and Future Directions" section.

Pages 11-12, Lines 231-238 and 259-263

ECG tracings: in a review of ECG abnormalities, actual tracings would enhance the manuscript. Specifically the tracings would be of more unusual or unique findings in acute pulmonary embolism such as S1Q3T3, clockwise rotation or low voltage findings. One tracing may be in the manuscript with the others in the supplement.

We agree that including actual ECG tracings would significantly enhance the manuscript's clinical relevance and provide valuable visual examples of the abnormalities discussed. We will incorporate representative ECG tracings, focusing on the more unique findings associated with acute PE, including S1Q3T3, T wave inversions in leads V1-V2, and right bundle branch block, as suggested. The other ECG patterns mentioned in our study are considered common and non-specific. Hence, the remaining ECG tracings could be found in the literature elsewhere and would not be presented in this manuscript.

Figure 3

Reviewer 2 Report

Comments and Suggestions for Authors

The authors present a study that conducts a systematic review and meta-analysis of 24 studies involving 7,467 patients with acute pulmonary embolism (APE), focusing on the prevalence of electrocardiographic (ECG) alterations. The most common findings were sinus tachycardia (31%), rotary rotation (28%), T-wave inversion in V1-V3 (18%), S1Q3T3 pattern (15%), and right bundle branch block (14%). Although the results are consistent with previous literature on right ventricular overload in APE, the high heterogeneity (I² > 95%) directly limits the clinical applicability of the data. This finding of the data is provided by the authors themselves.

Despite the value of the ECG as an accessible and low-cost tool, the study reaffirms its isolated diagnostic limitations. International guidelines such as those of the European Society of Cardiology (ESC, 2019) reinforce that the ECG should not replace imaging tests, such as computed tomography pulmonary angiography (CTPA), which is recommended as the gold standard. The World Health Organization (WHO) and cardiology societies, such as the American Heart Association (AHA) and the American College of Cardiology (ACC), also warn of the risk of delays in diagnosis if there is excessive reliance on the ECG.

The initial diagnosis of pulmonary embolism (PE) should begin with a structured clinical evaluation, using pre-test probability scores, such as the Wells score or the Geneva score, followed by D-dimer measurement in cases of low or moderate probability. Patients with signs of hemodynamic instability should immediately undergo computed tomography pulmonary angiography (CTPA), the gold standard exam for diagnostic confirmation. In settings with limited resources or contraindications to CTPA, ventilation-perfusion (V/Q) scintigraphy and bedside echocardiography can be used. Chest radiography and ECG should be performed routinely, although they have low specificity, and can help exclude differential diagnoses. An early and systematic approach reduces the risk of fatal complications. The 2019 ESC guidelines reinforce the role of risk stratification in the initial management of PE.

The ECG is a useful tool in patient monitoring conditions and should be considered in the evolution of a clinical picture. ECG variations are an indicator of the onset of unstable hemodynamic conditions.

The S1Q3T3 pattern, traditionally seen as indicative of APE, showed low sensitivity, which agrees with studies such as that of Geibel et al., which question its isolated usefulness. On the other hand, T-wave inversion in V1–V3 appears to have greater diagnostic sensitivity, being indicated in recent reviews as a marker of severity. The presence of RBBB, in turn, has been correlated in previous studies with higher early mortality, indicating potential prognostic use.

The article is methodologically robust and very well structured, varying specific variations and findings of ECG disturbances. However, the study suffers from variability in the definitions of ECG patterns and collection times. The lack of standardization of ECG interpretation criteria, highlighted by the authors themselves, limits the reproducibility of the findings. In addition, the absence of analyses by APE severity, sex or age prevents more refined clinical inferences.

Although the bibliographic data provide support for the use of ECG as a complementary tool, there is an urgent need for multicenter prospective studies with uniform protocols. ECG can be useful for clinical suspicion in low complexity settings but should not be used alone for diagnostic or therapeutic decision-making, as recommended by the ESC. Its prognostic potential, especially in patterns such as RBBB and inverse T, deserves further exploration in future clinical studies.

Author Response

Response to Reviewers

Date: 17 June 2025

Title: Prevalence of Electrocardiographic Abnormalities in Patients with Acute Pulmonary Embolism: A Systematic Review and Meta-analysis

Reviewer 2:

Comments

Responses

Correction on/ Mentioned on

The authors present a study that conducts a systematic review and meta-analysis of 24 studies involving 7,467 patients with acute pulmonary embolism (APE), focusing on the prevalence of electrocardiographic (ECG) alterations. The most common findings were sinus tachycardia (31%), rotary rotation (28%), T-wave inversion in V1-V3 (18%), S1Q3T3 pattern (15%), and right bundle branch block (14%). Although the results are consistent with previous literature on right ventricular overload in APE, the high heterogeneity (I² > 95%) directly limits the clinical applicability of the data. This finding of the data is provided by the authors themselves.

We appreciate the accurate summary of our main findings and the acknowledgment of the observed high heterogeneity. We fully agree with the reviewer that the substantial heterogeneity (I² > 95%) across the included studies is a significant limitation impacting the direct clinical applicability of individual pooled prevalence estimates. As noted by the reviewer, we have explicitly highlighted this limitation in our Abstract and Discussion sections, emphasizing the need for larger, higher-quality studies with standardized ECG protocols to better understand ECG's diagnostic and prognostic role in PE.

Page 1, Lines 30-32; Page 11 Lines 239-241

Despite the value of the ECG as an accessible and low-cost tool, the study reaffirms its isolated diagnostic limitations. International guidelines such as those of the European Society of Cardiology (ESC, 2019) reinforce that the ECG should not replace imaging tests, such as computed tomography pulmonary angiography (CTPA), which is recommended as the gold standard. The World Health Organization (WHO) and cardiology societies, such as the American Heart Association (AHA) and the American College of Cardiology (ACC), also warn of the risk of delays in diagnosis if there is excessive reliance on the ECG.

We completely agree with the reviewer's assertion regarding the limitations of ECG as an isolated diagnostic tool for acute PE. Our manuscript consistently emphasizes that while ECG is a widely available, rapid, and non-invasive tool that can provide diagnostic clues and raise clinical suspicion, it "lacks the sensitivity and specificity to serve as an isolated definitive diagnostic test" and should not replace imaging modalities like CTPA, which remains the gold standard for diagnosis. We have cited the ESC 2019 guidelines in our Introduction and acknowledge the importance of a structured diagnostic approach. Our discussion underscores that ECG's value lies in "raising clinical suspicion in appropriate contexts" rather than serving as a definitive diagnostic test on its own.

Page 11, Lines 245-257

The initial diagnosis of pulmonary embolism (PE) should begin with a structured clinical evaluation, using pre-test probability scores, such as the Wells score or the Geneva score, followed by D-dimer measurement in cases of low or moderate probability. Patients with signs of hemodynamic instability should immediately undergo computed tomography pulmonary angiography (CTPA), the gold standard exam for diagnostic confirmation. In settings with limited resources or contraindications to CTPA, ventilation-perfusion (V/Q) scintigraphy and bedside echocardiography can be used. Chest radiography and ECG should be performed routinely, although they have low specificity, and can help exclude differential diagnoses. An early and systematic approach reduces the risk of fatal complications. The 2019 ESC guidelines reinforce the role of risk stratification in the initial management of PE.

We fully endorse the comprehensive diagnostic algorithm outlined by the reviewer, which aligns with international guidelines for the diagnosis and management of acute PE. Our manuscript references the Wells and revised Geneva scores in the context of sinus tachycardia's inclusion in clinical prediction rules and acknowledges CTPA as the diagnostic gold standard. While our study specifically focused on ECG abnormalities, we agree that ECG should be interpreted within the broader context of a structured clinical evaluation, including pre-test probability assessment, D-dimer testing, and appropriate imaging (CTPA, V/Q scan, echocardiography) and risk stratification. We have reinforced this overarching diagnostic principle in our Discussion section to provide a more complete picture of the ECG's role in the diagnostic pathway.

Page 11, Lines 246-257

The ECG is a useful tool in patient monitoring conditions and should be considered in the evolution of a clinical picture. ECG variations are an indicator of the onset of unstable hemodynamic conditions.

We strongly agree with this important point. Our manuscript highlights that certain ECG findings "may offer valuable clinical clues, particularly when advanced imaging is delayed or unavailable,” and discusses how changes like transient RBBB can suggest a direct hemodynamic effect. In our "Clinical Implications and Future Directions" section, we advocate for future research to "evaluate the temporal evolution of ECG changes in PE and their prognostic significance across varying clinical severities." We have further emphasized the role of ECG as a valuable tool for patient monitoring and as an indicator of evolving hemodynamic status in the Discussion section.

Page 2, Lines 53-54; Pages 11-12, Lines 259-263

The S1Q3T3 pattern, traditionally seen as indicative of APE, showed low sensitivity, which agrees with studies such as that of Geibel et al., which question its isolated usefulness. On the other hand, T-wave inversion in V1–V3 appears to have greater diagnostic sensitivity, being indicated in recent reviews as a marker of severity. The presence of RBBB, in turn, has been correlated in previous studies with higher early mortality, indicating potential prognostic use.

We appreciate the reviewer's astute observations, which are consistent with our findings and discussions. Our manuscript explicitly states that the S1Q3T3 pattern, while traditionally considered a hallmark, was found to align with its "high specificity but low sensitivity". We also discuss that "T-wave inversion may offer higher sensitivity and moderate diagnostic accuracy, underscoring its potential diagnostic value in combination with other findings". Furthermore, we emphasize that "accumulating evidence suggests that certain ECG features—particularly RBBB and T-wave inversions—may also carry prognostic significance in risk stratification tools, potentially informing decisions regarding thrombolysis or intensive monitoring". We ensure these points are clearly articulated and, where possible, further strengthened in our Discussion section.

Page 10, Lines 187-189 and 199-203

The article is methodologically robust and very well structured, varying specific variations and findings of ECG disturbances. However, the study suffers from variability in the definitions of ECG patterns and collection times. The lack of standardization of ECG interpretation criteria, highlighted by the authors themselves, limits the reproducibility of the findings. In addition, the absence of analyses by APE severity, sex or age prevents more refined clinical inferences.

We are grateful for the recognition of our methodological robustness and structured approach. We wholeheartedly agree with the reviewer's points regarding the limitations arising from variability in ECG pattern definitions, collection times, and the lack of standardized interpretation criteria across primary studies. As highlighted by the reviewer, we have detailed these limitations in our manuscript, stating that "Most concerns arose from unclear sampling methods, sample size, absence of consecutive patient inclusion, and lack of standardized ECG interpretation." We also explicitly mention that "incomplete reporting in the primary studies precluded subgroup analyses by sex, age, PE severity, or comorbidities. Additionally, the lack of standardized ECG acquisition protocols and variability in outcome definitions may have affected study comparability." These limitations underscore our call for future higher-quality studies with standardized protocols to enable more refined clinical inferences and enhance reproducibility.

Page 3, Lines 137-139; Page 11, Lines 239-241

Although the bibliographic data provide support for the use of ECG as a complementary tool, there is an urgent need for multicenter prospective studies with uniform protocols. ECG can be useful for clinical suspicion in low complexity settings but should not be used alone for diagnostic or therapeutic decision-making, as recommended by the ESC. Its prognostic potential, especially in patterns such as RBBB and inverse T, deserves further exploration in future clinical studies.

We strongly agree with the reviewer's concluding remarks. Our manuscript's "Clinical Implications and Future Directions" and "Conclusions" sections advocate precisely for this: "the need for standardized ECG interpretation and larger, high-quality studies remains critical. Future research should focus on the prognostic significance of these ECG findings and their integration into risk stratification models to enhance patient management." We reiterate that ECG serves as a valuable "complementary tool" for raising clinical suspicion, particularly in resource-limited settings, but not for isolated diagnostic or therapeutic decision-making. We have emphasized the urgent need for multicenter prospective studies with uniform protocols to fully explore the diagnostic and prognostic potential of ECG patterns like RBBB and T-wave inversions, ensuring their robust integration into clinical guidelines.

Page 12, Lines 270-277
